# Classifying Malignancy in Prostate Glandular Structures from Biopsy Scans with Deep Learning

**DOI:** 10.3390/cancers15082335

**Published:** 2023-04-17

**Authors:** Ryan Fogarty, Dmitry Goldgof, Lawrence Hall, Alex Lopez, Joseph Johnson, Manoj Gadara, Radka Stoyanova, Sanoj Punnen, Alan Pollack, Julio Pow-Sang, Yoganand Balagurunathan

**Affiliations:** 1Department of Machine Learning, H. Lee Moffitt Cancer Center, Tampa, FL 33612, USA; 2Department of Computer Science and Engineering, University of South Florida, Tampa, FL 33620, USA; 3Tissue Core Facility, H. Lee Moffitt Cancer Center, Tampa, FL 33612, USA; 4Analytic Microscopy Core Facility, H. Lee Moffitt Cancer Center, Tampa, FL 33612, USA; 5Anatomic Pathology Division, H. Lee Moffitt Cancer Center, Tampa, FL 33612, USA; 6Quest Diagnostics, Tampa, FL 33612, USA; 7Department of Radiation Oncology, University of Miami Miller School of Medicine, Miami, FL 33136, USA; 8Desai Sethi Urology Institute, University of Miami Miller School of Medicine, Miami, FL 33136, USA; 9Genitourinary Cancers, H. Lee Moffitt Cancer Center, Tampa, FL 33612, USA

**Keywords:** prostate, Gleason cancer grading, pathology, uropathology, whole-slide image, ISUP grade, Gleason score, deep learning, convolutional neural network, transfer learning

## Abstract

**Simple Summary:**

In recent years, the prostate cancer histopathological description proposed by Gleason has emerged as a universal standard used for disease diagnosis and progression. Recently, a grading scheme on a point scale is based on Gleason patterns. Current scores are highly dependent on the expert urinary histopathologist and show a high level of variability among experts. To aid the clinician, we have developed deep learning models that provide a decision aid in identifying the primary cancer grade (dominant Gleason pattern).

**Abstract:**

Histopathological classification in prostate cancer remains a challenge with high dependence on the expert practitioner. We develop a deep learning (DL) model to identify the most prominent Gleason pattern in a highly curated data cohort and validate it on an independent dataset. The histology images are partitioned in tiles (14,509) and are curated by an expert to identify individual glandular structures with assigned primary Gleason pattern grades. We use transfer learning and fine-tuning approaches to compare several deep neural network architectures that are trained on a corpus of camera images (ImageNet) and tuned with histology examples to be context appropriate for histopathological discrimination with small samples. In our study, the best DL network is able to discriminate cancer grade (GS3/4) from benign with an accuracy of 91%, F_1_-score of 0.91 and AUC 0.96 in a baseline test (52 patients), while the cancer grade discrimination of the GS3 from GS4 had an accuracy of 68% and AUC of 0.71 (40 patients).

## 1. Introduction

Prostate cancer is a neoplasm in the prostate gland, most often epithelial in origin, with over 95% of adenocarcinoma subtype. The neoplasms are classified from different grades of aggressiveness using Gleason patterns 1–5, then combined into a Gleason score (dominant + subdominant Gleason patterns) detailed below [1,2]. Standard diagnosis requires a fine needle biopsy of the gland where the histology is assessed from hematoxylin and eosin (H&E)-stained tissue sections by an expert genitourinary pathologist [1]. The prostate adenocarcinomas histopathology displays an abnormal architectural glandular pattern with a very high degree of benign epithelial–stromal relationships. Most widely used Gleason scoring patterns were adopted by the International Society for Urological Pathology (ISUP) proposed in 2014, later adopted by the World Health Organization (WHO) in 2016 [3]. The patterns are described by a modified Gleason grading that shows five distinct patterns with direct relationships to cancer invasiveness, which were conceived purely based on clinical outcomes [4]. The pattern spans from single, separated well-formed glands in Gleason pattern 1, ISUP grade group 1 (GS 3 + 3), to stromal infiltration in Gleason patterns 4 to 5 (ISUP grade groups 2 to 5) [4]. The cancer types have relied more on the epithelial–stromal architecture than any other clinical grade based classification to describe disease aggressiveness with direct relation to the clinical outcome [5]. Use of the ISUP scoring scheme has helped to reduce the scoring range, but this expert-based standard has significant intra- and inter-variability among genitourinary pathologists and clinical centers and results in care differences among patients [6,7,8]. In a recent report, concordance rates between two observers for primary and secondary Gleason patterns were 63.96% (κ = 0.34) and 63.45% (κ = 0.37), respectively, while Gleason grades was at 57.9% (κ = 0.39) [9]. This does not get better with diagnoses around the globe; concordance ranged from 0.44 to 0.49, while urological pathologists showed moderate improvement to 0.68 [10].

Development in the last decade has seen promise in using machine learning (ML) and deep learning (DL) tools to improve diagnostic variability and provide a decision support system (DSS) to aid the pathologist and improve quality of care or treatment response [11,12,13,14,15,16]. A potential implementation of a DSS is shown in Figure 1; this study concentrates on the decision classifier. A complete implementation of a DSS will include many preprocessing steps such as region extraction, which were supported through preprocessing tiles as detailed in Section 2. Feature extraction and composition through explicit means (not implicitly derived through CNN layers) such as radiomic features and feature composition is a fertile area for improvement [17,18] but not a focus of this study. Our minimalist approach is covered in the proceeding. Recent deep learning techniques with convolutional neural networks (CNN) have shown tremendous promise in extracting non-human visible salient features from diagnostic pathological images [19,20,21,22,23,24]. Our results demonstrate that these diagnostic clues are available in local glandular structures or small patches of prostate biopsy whole-slide images (WSI). Due to the subtle nature of the features in histopathological images coupled with limited sample sizes, generalization of the model continues to be a challenge across medical centers and sources [25].

Recently, others have shown use of deep network’s ability to identify cancer grades, progression and outcome using medical imaging (CT/MRI) utilizing various data augmentation methods and transfer learning (TL) approaches [14,15,26,27,28,29]. Improving classification by transfer learning from large data sources such as ImageNet has been a standard approach for many years for many medical imaging modalities [30]. However, more recent studies have shown that TL may be limited to learning patterns in small sample sets [31,32].

Our research goal is to develop a deep learning model to discriminate Gleason grades in whole-slide H&E images. In this study, our focus is to accurately identify aggressive primary Gleason patterns in annotated image patches. To enable proper network model training, the WSIs of prostate needle biopsies were manually annotated following the pathologist-assigned Gleason pattern scored at the gland level. We focused on evaluating several deep learning model architectures based on CNNs (EfficientNet [33], ResNet [34]), and the Visual Geometry Group (VGG-16 and VGG-19 [35]) networks. Each has advantages, but in this domain, VGGs showed the best performance by a significant margin. Each of these networks were trained using transfer learning, with a degree of fine tuning on feature weights to adapt the network to the histopathological classification as previously used in other studies [36].

There have been few attempts in the past to discern the gland patterns. This study, to our knowledge, is one of the first to create a large cohort of manual gland level scoring (over 14k glands, see Table 1) and use deep learning (DL)-based models to discriminate the Gleason patterns. We believe these results form a baseline comparison for the primary patterns at this level of granularity. In comparison, most other studies have used DL-based classification algorithms to discern Gleason grades or scores (primary + secondary) at the whole-slide level, containing multiple glands that represent primary and secondary patterns, most often in an unbalanced proportion [16,21]. Ström et al. created an ensemble of 30 Inception V3 nets to score individual patches as benign or a Gleason pattern and pass the results through a boosted decision tree to inform an overall Gleason (ISUP) grade [22]. Pinckaers et al. demonstrated a novel streaming CNN (based on ResNet-34) to process entire biopsy scans’ centered, cropped patches to detect malignancy status (cancer from benign) at the whole-slide level [21]; this method was compared against two baseline approaches from Companella et al. and Bulten et al. Companella et al. applied multi-instance learning (MIL) with a recurrent neural network (RNN) to predict a combined score from patches classified with a ResNet-34 [37]. Bulten et al. used an extended U-Net to predict Gleason patterns at the pixel level and to determine the proportions of malignancy to inform an overall grade decision [19]. It is to be noted that most prior work has focused on cancer status discrimination at the slide level assessing multiple pattern (primary and secondary) scores. In contrast, our proposed work focuses on a discriminating Gleason pattern (primary) at the patch level, limiting the reader variability. Hence, our performance can be qualitatively compared with prior research. In this work, we demonstrate the use of a VGG CNN model for classifying small patches (or individual, variably sized glands) in a WSI obtained from a core needle prostate biopsy. Additionally, we contrast the model performance on two diverse datasets independently obtained.

In the following Section 2, the studied datasets are discussed, as well as the techniques used in this study, details on the models (DL architectures), training and tuning techniques and data processing. In Section 3, we show the results for our findings on Gleason pattern discrimination (benign versus cancer (all grades); GS3 versus GS4) at the prostate gland level. Section 4 provides further discussion of the results and use of a decision support system to improve clinical diagnosis. Finally, in Section 5, we summarize our findings.

## 2. Materials and Methods

### 2.1. Data Cohorts

Our study used two retrospectively curated data cohorts of prostate cancer patients’ biopsies obtained at two different cancer centers. Patients’ data were retrospectively obtained using the respective clinical centers’ research protocols. The data were de-identified for research use, obtained after an Institutional Review Board (IRB) review of our research project, and the patients waived their informed consent rights for retrospective research usage. The first cohort is a dataset obtained from the University of Miami (UM) and curated at the H Lee Moffitt Cancer Center (MCC) and will be referred to as the UM/MCC data cohort. The second cohort was derived from the Kaggle PANDA histopathology open challenge; the de-identified patient data with Gleason grades were provided by Radboud University, Nijmegen, the Netherlands as part of their effort to promote open science—available on the National Institute of Health’s Cancer Imaging Archive website (https://www.cancerimagingarchive.net/ (accessed 24 February 2023). The patient data from both sources were completely anonymized with no treatment details or outcomes provided.

#### 2.1.1. Gland Level Patient Data Cohort

We obtained digitized whole-slide histopathology with 20× magnification scanned on an Olympus VS120 scanner (Olympus Life Sciences, Inc., Tokyo, Japan). These images were imported into the Visiopharm^®^ digital pathology software, and gland regions were manually delineated and annotated by our research urinary pathologists (AL and MG) with over 15 years and 9 years of clinical experience in prostate histopathology scoring, respectively. The glands were scored on benign, GS3, GS4 or GS5 levels.

The study used 52 patients, 150 WSIs and 14,509 glands; the data will be referenced as UM/MCC data (see Table 1). An independent cohort was assembled from the Kaggle PANDA’s challenge used as a training/validation cohort (at a 90/10 ratio), with 24,800 patches with glands having the same primary pattern (see Table 2).

To train or test on WSIs, (relatively) small image patches were created for each of the labeled glands (dominant Gleason pattern). These specific glands were identified and converted from vector files in the Visiopharm^®^ MLD format into segmentation masks. These masks were then used to extract individual image patches using several bounding box techniques from the WSIs, including fixed-size bounding boxes, squared bounding boxes and tight bounding boxes. The latter two techniques resulted in image patches of various sizes, which required subsequent resizing or resampling techniques to be used by the CNN ML (details deferred to Section 2.2.1).

Figure 2 shows several patches with tight bounding boxes for the classification levels benign, GS3, or GS4, respectively. Note that the images of glands shown below are within 15% of 120 × 120, 240 × 240, 360 × 360 and 720 × 720 pixels. The aspect ratios of the sample set’s width and height has much higher variability than the samples shown.

Due to limited dataset size, we estimated the discriminators’ performance using the Monte Carlo cross-validation (MCCV) technique and reported the ensemble statistics [38], which will be shown in Section 3. Due to extensive time required for network training on the computational resource, retraining of the network was avoided and re-sampling of the outcome was resorted to.

#### 2.1.2. PANDA Radboud Data Cohort

We curated the large open-source prostate pathology data cohort shared as a part of the prostate cancer grade assessment (PANDA) challenge organized through the Kaggle open challenge platform [16]. The PANDA training set had expert-provided annotations; it was composed of two separate patient sources: the Karolinska Institute and Radboud University. In our study, we used images from the Radboud collection because of the fidelity of the labeled segmentation masks that overlapped with the cases in the UM/MCC dataset.

The PANDA Radboud dataset, scanned from needle biopsy slides, was synthesized into a set of data patches from the Kaggle-provided data. The partial WSI images were downsampled by 2× then Otsu binarized to isolate foreground and background. Fixed-sized patches were extracted from the foreground (biopsy image) area by sliding a window (400 × 400) over the label mask (where foreground was identified) and accepting patch areas that contained an appropriate density of segmented (labeled) data for some target Gleason score. The degree of window overlap p_ω_ was adjusted until each WSI sample produced at least 20 patches (overlap p_ω_ starting from 0.5 and adjusted as high as 0.8 if enough samples were not generated per image). Several thresholds were tuned to generate an approximately equal set of patches for each Gleason pattern level by testing each label mask pixel (x_i_) for the ratio of Gleason–label mask p_α_ (nominally ≥ 0.1) that was not identified as background or stroma and the purity of label p_β_ (0.95) at the targeted Gleason level. The cut points for the quality filters were heuristically fixed at these levels.
$$p_{\alpha }=0.1\;p_{\beta }=0.95\;N=width*height\;l_{epithelium}=2\;M=\sum_{i=1}^{N}x_{i}\geq l_{epithelium}\;l_{target}=GS;GS\in \{3,4,or\;5\}\;T=\sum_{i=1}^{N}x_{i}\equiv l_{target}\;accept=\frac{M}{N}\geq p_{\alpha }\;\&\;\frac{T}{N}\geq p_{\beta }$$

Once these patches were generated for each Radboud image file, the patches were rank sorted by the proportion of epithelial/malignant label mask coverage *M/N*, and the 20 patches with highest ratio of label were added to a synthesized set of patches for Gleason levels benign, GS3, GS4 and GS5. If, after adjusting the sliding window overlap as high as p_ω_ = 0.8, an image was still unable to produce 20 patches, it was excluded from the training set. The PANDA Radboud dataset may include multiple glands per patch, which is different than the UM/MCC data but is sufficient as a pretraining dataset for distinguishing Gleason patterns. In Figure 3, patches and their corresponding Radboud masks are shown for Gleason levels benign, GS3, GS4 and GS5.

In generating this dataset, we made the training and validation/test cohorts as uniform as possible for the Gleason pattern; patches for benign were only pulled from images with clinical diagnosis benign, ISUP grade 1 (3 + 3) for GS3 pattern and grade 4 (4 + 4) for GS4 pattern. For Gleason pattern 5, very few images were graded as 5 + 5; therefore, patches were drawn from 4 + 5, 5 + 4 and 5 + 5 samples. The patch quality metrics, epithelial/glandular density p_α_ (nominally ≥ 0.1) and purity of label p_β_ = 0.95 ensured that each patch was appropriate for the primary Gleason grade. Note that this procedure was not required for the UM/MCC dataset since patches were extracted from each labeled gland. Table 2 shows the resultant dataset with 310 images and the corresponding 6200 patches per each Gleason level. Most of the clinical data such as subject identification was excluded from the Kaggle PANDA collection. The patient data was completely anonymized with no outcome or treatment data provided.

### 2.2. Image Preprocessing

#### 2.2.1. Sample-Mix

A necessary step in preprocessing the data was to ensure the image patches were identical in size prior to being processed through the DL CNN. There have been many resizing techniques that were tried previously such as in [39,40], cropping to fixed size areas (or loose bounding boxes) around the areas of interest, such as the glands. In our study, we adopted a *sample-mix* approach, which was inspired by other image mixing techniques such as in [41,42,43]. The approach preserves the scale and the aspect ratio of textural features; see sample in Figure 4. The tiled approach allows smaller and larger images to be adjusted to the same size, preserving their original textural characteristics, immaterial of the gland size (small or large). It is possible to construct sample-mixed patches for any dimension and rank, where rank is the number of tiles sampled along each axis. The examples show a target dimension of 300 × 300 pixels and rank of three tiles (along the horizontal and vertical), requiring the sampled tiles to be 100 × 100 (or 1/9 of the target size).

The sample-mix methodology, a resize strategy, was used when image data sources were variably sized and needed to be resized to train the network model. We used this strategy in the UM/MCC training cohort, where the extracted patches with tight bounding boxes were of different sizes. For images that were smaller in the horizontal or vertical direction than the sample tile (100 × 100), those samples were simply removed from the dataset as a preprocessing step. When data elimination is not desired, the algorithm can automatically generate a sample mix with smaller sample tiles.

#### 2.2.2. Standardization

The technique of staining hematoxylin and eosin (H&E), respectively, provides pathologists with functional and morphological details at the cellular level. It is well-recognized that even after over a century of its usage, there are many variables such as the stain protocol, dye quality and dye age that are uncontrolled factors causing inter-laboratory variability resulting in visual differences in slide appearance (color and intensity) [44,45].

The staining of images across sources is often inconsistent, and varying degrees of chemical application may result in significantly different color saturation [46]. In Figure 5, three partial views of WSIs with clearly varied colors are shown; the first two are from the same UM/MCC cohort, and the third on the right is from PANDA Radboud data. It is possible that these shade differences between the cohorts dampens generalization of the DL models.

To mitigate these wide variations, the image patches were normalized using z-score standardization, where color channels are re-centered to zero mean and unit standard deviation. This technique is the most often used approach for training on large image datasets such as ImageNet [27]. As a result of applying transfer learning starting with ImageNet weights, standardization is a necessary preprocessing step to ensure proper feature extraction through the CNN layers. Unfortunately, this transformation step may not always improve a model’s reproducibility in histopathology images. It has been reported that most H&E-stained image intensity follows a bi- or tri-modal distribution [47,48,49]. The standardization followed in most learning methods uses linear scaling that may not compensate for the distributional spread, which would be an additional source of alteration in the model training.

#### 2.2.3. Eliminating Outliers

Outlier detection (OD) is an important step in maximizing performance of an ML algorithm [50]. To remove outliers in the dataset, we performed the random sampling and consensus (RANSAC) method as has been applied in regression problems [51]. In a classification problem, we remove samples that never or very seldom classify correctly after an initial training of the DL models. The technique is similar to a histogram-based OD in which outliers are removed based on a threshold rule, classically as a distance away from the 25% and 75% quartile, normalized by inter-quartile range or IQR. The RANSAC technique requires multiple models to be trained on the data (as is a natural approach when performing CV experiments); inference is then performed on the entire sample set for each model, and a consensus of each sample’s performance is determined. While performing the test, the samples that never classify correctly were removed from the training set. To preserve the integrity of the performance metrics, RANSACed outliers are removed from future training sets, but validation data are not altered.

In determining the outliers, the consensus scores are drawn by inferencing the training data with the models derived using the CV folds. As a result, the sample models will have seen the data many times, and hence, low consensus scores of no correct classification or one correct classification out of multiple models applied (e.g., 0/20 and 1/20) imply likely outliers. This technique was applied at least once for the binary DL classifiers shown in Section 3.

In Figure 6, the RANSAC consensus scores are shown as stacked histograms when training GS3 versus GS4 DL models. The colors in the vertical bars and the legend along the bottom of the chart represent different consensus scores in 20 models; thus, the worst outliers score 0/20 times, and the best performers score 20/20 times. The stacked histograms add up to the total number of subjects in our training set, and as can be seen, with each iteration, several low-performing samples are eliminated from the set. Additionally, this plot demonstrates how with each iteration, the remaining training samples see a gradual improvement in the consensus score. Other unsupervised or semi-supervised OD techniques are under consideration for future work [52,53].

#### 2.2.4. Balancing Data

The GS3 versus GS4 classification experiment was largely unbalanced; in the UM/MCC cohort, the GS3 majority class was twice as large as the GS4 minority class. To ensure that the machine learners did not simply prefer the majority class, in all experiments, the data was balanced. To train DL networks, we used bootstrapping to dynamically balance both training and validation sets [54]. A custom TensorFlow iterator was created to ensure data was balanced on every batch training update. Performance statistics were estimated using the bootstrap technique.

### 2.3. Deep Learning

A convolutional neural network (CNN) utilizing transfer learning from very large datasets such as ImageNet shows promise for classification problems. Networks with lower inductive bias are expected to outperform CNN architectures as more and larger datasets become available (such as those from Kaggle PANDA), through knowledge distillation and improved architectures that optimize generalized learning [55,56,57].

Most DL models that are studied with CNN feature layers combine with a binary or multiclass dense classification layer. Common techniques such as dropout, pooling and batch normalization were used between layers to improve performance [58,59,60]. The fully connected classification layer has a 32-node layer to aggregate features from the CNN, followed by as many neurons as are required for the classification task (1 for binary classification, or 3+ for multi-classifier), as shown in Figure 7.

#### 2.3.1. Optimization Technique

Research has shown that adjusting the learning rate in a cyclic fashion can help to escape local minima and saddle points [61]. The learning rate may be “shocked” or annealed by jumping back to a maximum on a periodic basis [62]. Our optimization strategy leveraged the cosine annealing technique. In this technique, the learning rate is adjusted from a maximum rate to a lower rate (perhaps one or two orders of magnitude smaller) and updated at each batch (partial training of epoch). Cosine-annealed training was used in this study, cycled every 29 epochs in our experiments. Additional hyper parameter values are shown in the Appendix A.

#### 2.3.2. Transfer Learning

Transfer learning is very effective at jumpstarting NN training, especially when data are limited. The UM/MCC cohort is relatively small; hence, we began training by initializing a VGG-16 network on ImageNet feature weights, a popular approach shortly following the original AlexNet [27,63]. We started by coarse tuning the fully connected (FC) classification layers with the UM/MCC data and followed by fine tuning the CNN feature layers and FC layers. We improved the results by first training on a larger Kaggle PANDA Radboud dataset (both a coarse tune followed by a fine tune to train the CNN feature weights), then fine tuning with data in our own UM/MCC cohort. The NN learner goes through four stages of learning, as shown in Figure 8. The technique of pretraining on one dataset and then tuning on another is a common and effective transfer-learning approach. A recent study corroborates this method when used for prostate pathology grading [64].

To create a more generalized model, we combined both our PANDA and UM/MCC datasets into one large training set and then trained our CNN to classify both sources. The combined dataset is also trained starting with ImageNet weights, first on the FC layers and eventually fine-tuning all weights in the FC and CNN layers.

### 2.4. Measuring Performance

Accurate measurement of ML performance poses a challenge when there are a limited number of subjects for model training. In the study, a randomized Monte Carlo cross-validation (MCCV) technique was used to estimate the network-based discriminators’ performance [38]. When applying MCCV, we performed a minimum of 20 folds, as recommended to ensure most data were tested since folds were sampled with replacements.

The following metrics were computed to evaluate deep networks’ classification performance: accuracy, sensitivity, specificity, precision, negative predicted value (NPV), F_1_-score and area under the receiver operating characteristic curve (AUC) [65]. Statistical metrics were computed using the Python scikit-learn metrics package (sklearn.metric) for DL experiments. These metrics were calculated across the folds following the recommendations described by Forman and Scholz [66]. The performance metrics are provided with 95% confidence intervals determined by the bootstrap estimation procedure [67].

### 2.5. Implementation Challenges

There were two main challenges for this study. The first major challenge was sample size and curation of a data cohort with gland-level labels in prostate histopathology. This work involved a clinical expert-driven manual gland scoring (semi-automatic) to create a pure cohort of about 14,000 labeled glands. The second challenge in this study was investigating the many state-of-the-art DL architectures, finding hyperparameters and tuning methods that resulted in optimized training with the diverse size of the gland-level patches. The computational resources required for model building in a timely manner posed a challenge.

## 3. Results

We used deep networks to perform binary classification tests to discriminate various grades of primary Gleason patterns at the glandular and small-tile level. These experiments compare benign versus GS3/4 (malignant) and finer grade levels, GS3 versus GS4 levels, as some may have clinical significance on the decision boundary of cancer progression and treatment.

We evaluated several types of deep networks for prostate histopathology classification, given the constraints of small sample sets. We found CNNs are well-studied with small sample dataset constraints, and multiple prior studies have shown stable performance with these constraints [68,69]. Several popular deep CNN architectures and their initial performances are detailed in the Appendix A. The VGG-16 and sample-mix technique proved to be the top performer and most practical, so all remaining tests include this combination. Appendix A shows performance for several alternative resizing techniques.

### Deep Network Performance

Results using the VGG16 CNN network trained on the PANDA Radboud, UM/MCC and combined dataset (PANDA + UM/MCC) follow. The models were built and trained in Python 3.8.10, Tensorflow/Keras 2.9.1 on the NVIDIA^®^ DGX™/A100 platform. Model files and samples of Python code are available for download at https://github.com/rfogarty/glandLevelGleasonClassification.git (accessed on 13 February 2023).

We found the best DL network trained on PANDA’s data (tile size of 400 × 400) showed exceptional results in discriminating cancer from benign with an AUC of 0.981 and AUC of 0.997 for discriminating cancer grades (GS3 vs. GS4), with other metrics shown in Table 3. The reported scores are cross-validation scores, and optimistic versus a holdout test set—the GS3 versus GS4 results are especially optimistic.

Table 4 summarizes performance of our VGG-16 DL architecture and establishes a baseline of performance for our UM/MCC dataset for the two binary problems studied. The networks were pretrained on ImageNet only (light blue columns) and pretrained on ImageNet followed by a pretraining on PANDA Radboud (darker blue columns). In this case, the benign versus GS3/4 binary classifier performed much better than the GS3 versus GS4, measuring better than 20% in almost all metrics. Although in both cases we scored better after a PANDA Radboud pretraining, the results of pretraining on PANDA only showed a marginal improvement. The scores for benign versus GS3/4 were measured using a conventional 10-fold CV, while the scores in GS3 versus GS4 classification were measured using a 20-fold Monte Carlo CV (because of very limited data size). Scores include 95% confidence intervals computed using the bootstrap method in parentheses.

Tests of cross-source generalization on our best PANDA-trained models and UM/MCC-trained models were poor, which demonstrates that the datasets are quite distinct. The best DL discriminator trained on PANDAs and tested on UM/MCC was able to differentiate cancer from benign (benign vs. GS3/4) with an AUC of 0.738. While the AUC drops to worse than random guessing (<0.5) for GS3 vs. GS4. In the alternate case, there was marginal performance for DL networks trained on UM/MCC and tested on PANDAs with an AUC of 0.522 for benign vs. GS3/4/5 and AUC of 0.692 for GS3 vs. GS4. Details of this experiment are shown in Appendix A. As shown below, we will significantly improve these results by training on a combined PANDA + UM/MCC dataset.

Finally, in Table 5, results are summarized for two networks that were simultaneously trained on a combined PANDA Radboud plus UM/MCC dataset. The columns in light green on the left are derived from the network that classifies benign versus malignancy (GS3/4), while the columns in dark green show results from a network that classifies a GS3 versus GS4 rating. The DL networks were configured to classify both source and Gleason pattern simultaneously, such as PANDA-GS3 or UMMCC-GS4, so both networks trained on four classes. These classes were then reduced to just a Gleason score for the inference decision (and comparison to patch label). Our AUC results on PANDA Radboud data, 0.988 for benign versus GS3/4 and 0.996 for GS3 versus GS4, demonstrate performance that equals the network trained solely on PANDA Radboud data. For our UM/MCC data, AUC is estimated at 0.963 for benign versus GS3/4 and 0.710 for GS3 versus GS4.

## 4. Discussion

DL methods have proven to be more effective in discriminating objects from different categories, exceeding human perception in the recent decade [27]. It is known that the DL methods’ performance drops in discriminating subjects that are sparse in their occurrence in the training sets. In our case, various quality factors affect our performance, which include inconsistent lighting conditions or stain quality, stain differences and generalizing across sources [70,71]. State-of-the-art approaches have ushered in techniques for much more complicated classification tasks, including Gleason scoring (GS) or ISUP Gleason grading of prostate pathology [72,73,74,75,76]. Classifying indolent from cancer grade based on H&E pathology with multiple glands is empirically a difficult problem—there is a subtle distinction between any neighboring patterns with a fuzzy discrimination boundary between the pattern scoring levels [8,77].

In our study, discrimination from benign/indolent versus cancer (GS3/4) grades shows excellent performance at the glandular level (AUC 0.96, Table 5). Grades of cancer discrimination GS3 versus GS4 performance was lower for the UM/MCC dataset (AUC 0.71, Table 5) but excellent for Kaggle PANDA Radboud data (whose patches may include multiple glands). Our results demonstrate that classifying individual glands at a Gleason level has acceptable performance and provides a basis to develop overall slide-level Gleason pattern scores. Grading individual glandular features has the advantage of increased fidelity in the decision process and provides supporting evidence for pathologists. Other studies have employed different approaches to the problem [78,79]. It is challenging to make a direct comparison with other findings; nevertheless, a comparison with recently published works follows. Singhal et al. show an accuracy of discriminating benign from malignancy of 0.85 on PANDA Radboud biopsy images [80], while our work shows an accuracy of 0.92 at the glandular level on our UMMCC data cohort and 0.96 on the PANDA Radboud data. Comparing with PANDA’s challenge data, in Bulten (2022) [16], a representative algorithm had validation metrics on sensitivity and specificity of tumor detection at 99.7% (98.1–99.7) and 92.9% (91.9–96.7), respectively, while our benign-versus-malignant classifier demonstrated Radboud scores of 94.4% (89.0–99.0) and 97.8% (95–1.0). In each of these comparisons, the results of our tests were tallied on single patches or glands, while the other studies compute results over the entire biopsy. See Table 5 for a summary of our results.

We used a cyclic learning rate to improve gradient descent, but the technique may also be used to choose an effective ensemble set [62]. Generally, we can train an ensemble set using a variety of methods to improve generalization and to lower variance [81]. The high variance reported on the VGG-16 DL network was contributed to by the small validation sets but also by not leveraging an ensemble technique.

The generalization of consistent ISUP grading or Gleason scoring across sources proves to be difficult if the network has not trained on that source or if the network is retrained on a new dataset (forgetting what it has previously learned). The Appendix A shows an example of catastrophic forgetting when testing PANDA data on a network fine-tuned on UM/MCC data, a common problem with machine learners [82]. Model training can be improved by continual learning, when integrating new sources, to ensure good cross-source generalization [83,84]. Alternatively, since datasets are relatively small within this domain, we successfully demonstrated combined-datasets training that integrated previously unseen sources, as shown in Table 5.

An orchestrated solution classifying glandular features and detection of additional features (such as density of nuclei and other recognized pathologic features) would exemplify a DSS that provides trust. Our contention is that ML and classifying entire WSIs is a useful aid, but DL CNNs, in particular, may make decisions that are not consistent with human observation and perception [12].

Using modern graphics processing units such as the NVIDIA^®^ A100, training on our combined prostate pathology cohorts require less than 1 h of compute time per DL model (roughly 17 h for 20 models of a 20-fold CV). A single inference decision on an image patch takes fractions of a second to process on any high-end commercial and consumer grade GPU device (a GPU accelerator is not needed for inference when deployed in the field), supporting rapid response and interrogation of data for histopathologists.

### Limitations and Future Improvements

A significant limitation to the approach is to classify higher grade patterns caused by limited samples in these grades. It is well-understood that a higher-grade Gleason pattern has progressively receding luminal regions and shows distinct morphological characteristics. Small sample size across the grades and varying glandular patterns make it difficult for the model to train and discriminate the patterns. We used the public cohort (PANDA Radboud) to find patches with Gleason pattern examples, but they may unavoidably contain multiple glands.

It is recognized that wide confidence bounds for some of our performance metrics could be attributed to smaller sample size. We believe using an ensemble technique, training on larger datasets and consensus scoring on WSIs will minimize the complexity of the DL model and improve the confidence bounds [85,86,87].

Future improvements to consider for generalization are consistent image preprocessing across sources, such as stain correction [14,88,89,90] and image resizing [91]. In future experiments, we intend to show improved generalization on unseen sources by first reducing discrepancies among data sources. Additionally, cross-source generalization could be improved by training an ensemble across many sources while ensuring that confounding factors are not contributing to shortcut learning—a generalized solution must learn from the wisdom of the masses to apply as a reference standard [92].

## 5. Conclusions

The baseline scores presented in this paper focus on discriminating primary Gleason patterns on individual glands or small patches of a prostate WSI. Our study demonstrates that a CNN DL model discriminates malignant patterns from benign tissue with a high level of accuracy. Furthermore, we were able to show validation of the findings in an independent larger-sized cohort (Kaggle PANDA Radboud data). Our work shows that classification of an indolent Gleason pattern from a clinically significant Gleason pattern shows impressive discrimination. We would need a larger sample cohort from diverse multi-centers to improve discrimination at the glandular level. Increasing the fidelity of an automated Gleason scoring scheme will provide a decision aid for clinical judgement.

It is well-recognized that clinical translation of pathological findings in the clinic would require improved region targeting. There are improved biopsy methods that show promise of improving tumor detection [93].

## Figures and Tables

**Figure 1 cancers-15-02335-f001:**
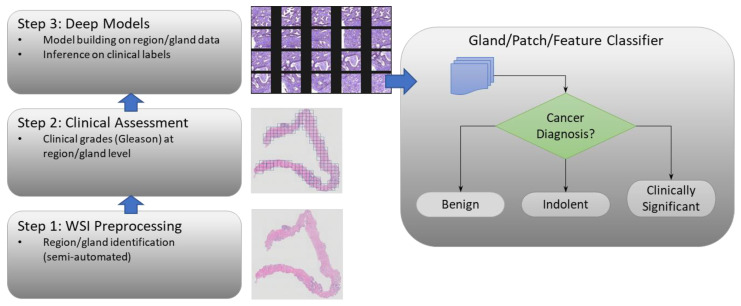
Prostate Cancer Decision Support System.

**Figure 2 cancers-15-02335-f002:**
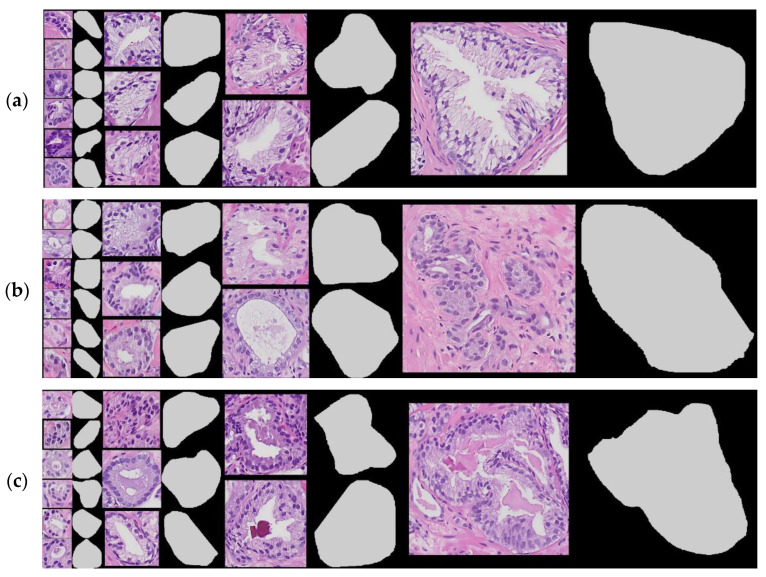
Sample selection of UM/MCC gland-level patches using the tight bounding-box technique paired with corresponding mask layer (on the right): (**a**) samples of benign, (**b**) samples of GS3, and (**c**) samples of GS4.

**Figure 3 cancers-15-02335-f003:**
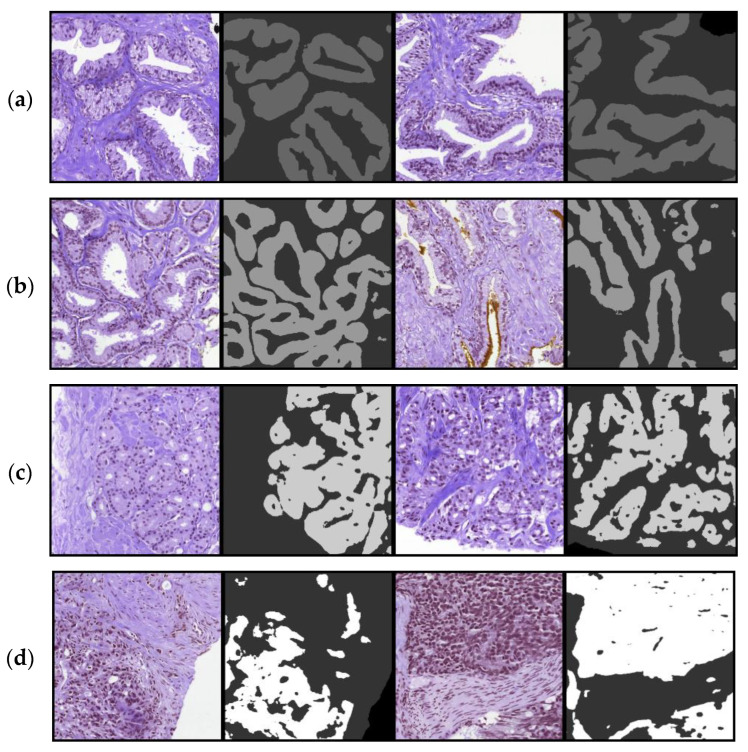
Sample selection of PANDA Radboud patches using the fixed-box technique paired alongside corresponding mask layer (on the right): (**a**) two samples of benign, (**b**) two samples of GS3, (**c**) two samples of GS4 and (**d**) two samples of GS5.

**Figure 4 cancers-15-02335-f004:**
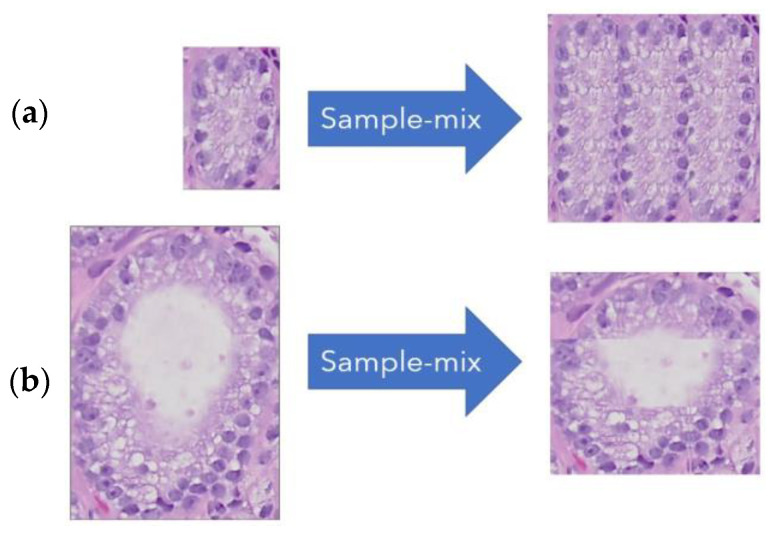
Sample-mix rescale examples: (**a**) 126 × 188 -> 300 × 300, (**b**) 290 × 417 -> 300 × 300.

**Figure 5 cancers-15-02335-f005:**
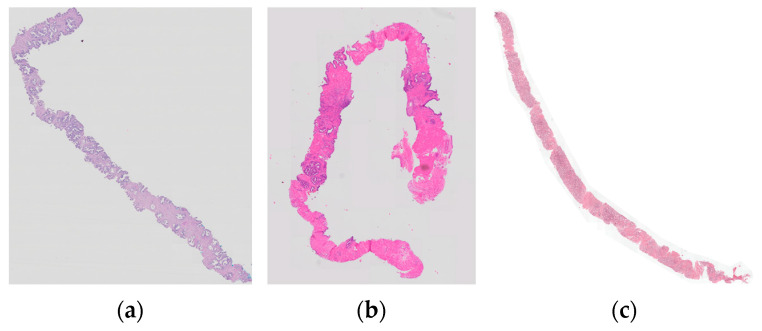
Examples to illustrate stain contrast differences among the samples in cohort: (**a**,**b**) UM/MCC dataset, (**c**) PANDA Radboud.

**Figure 6 cancers-15-02335-f006:**
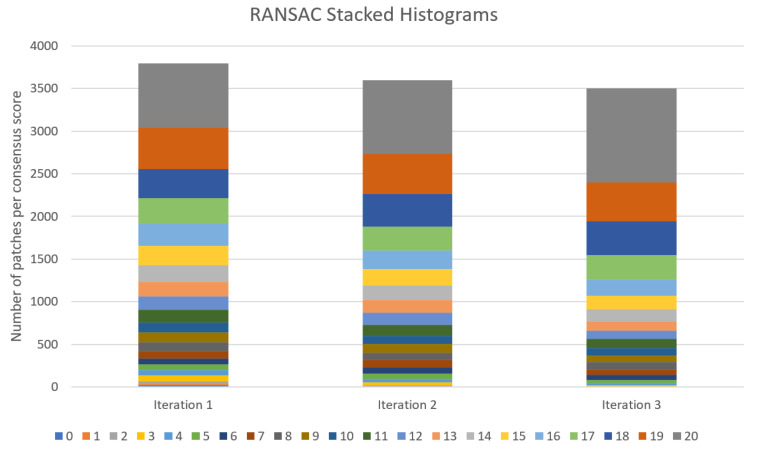
Three iterations of RANSAC histograms training GS3 vs. GS4 DL models.

**Figure 7 cancers-15-02335-f007:**
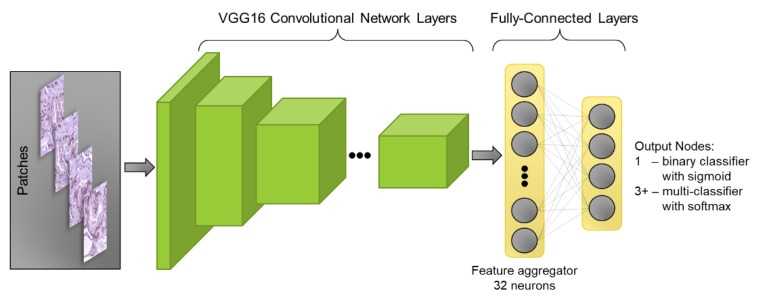
CNN feature layers, with dense classification layer.

**Figure 8 cancers-15-02335-f008:**
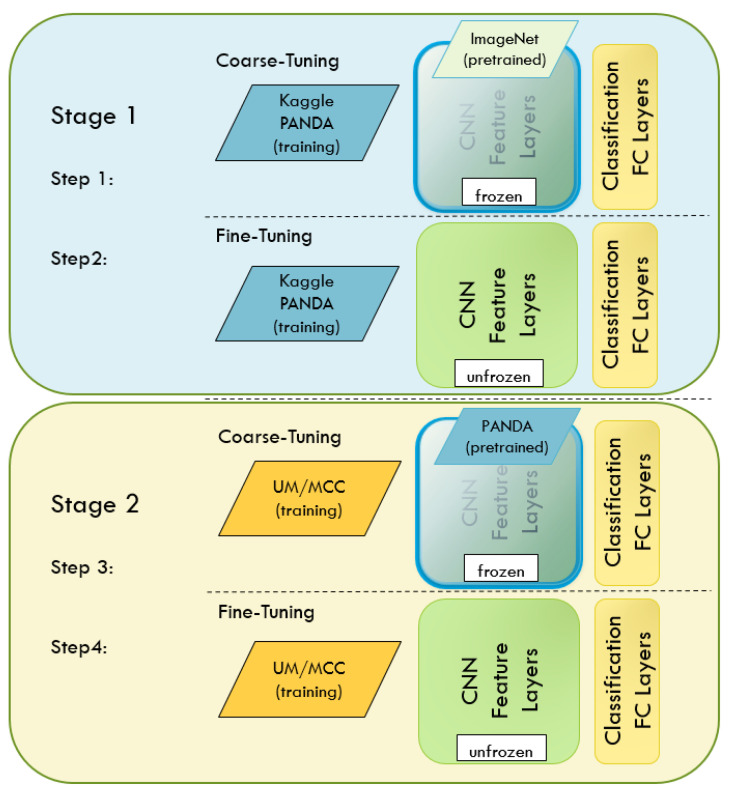
Four steps, transfer learning and fine tuning of NN.

**Table 1 cancers-15-02335-t001:** University of Miami/Moffitt Cancer Center Cohort.

	Total	Benign	GS3	GS4
Subjects	52	23	38	32
Whole-Slide Images	150	72	72	60
Labeled Glands	14509	6882	5143	2484

**Table 2 cancers-15-02335-t002:** Kaggle PANDA Radboud Synthesized Cohort.

	Total	Benign	GS3	GS4	GS5
Biopsy scans	1240	310	310	310	310
Patches	24,800	6200	6200	6200	6200

**Table 3 cancers-15-02335-t003:** PANDA Radboud classifier scores.

	Trained on PANDA Radboud	
	PANDA Radboud Benign vs. GS3/4/5	PANDA Radboud GS3 vs. GS4
Accuracy	0.941 (0.88, 0.98)	0.979 (0.95, 1.0)
Sensitivity	0.964 (0.92, 0.99)	0.980 (0.94, 1.0)
Specificity	0.920 (0.80, 0.98)	0.979(0.93, 1.0)
Precision	0.927 (0.83, 0.98)	0.979 (0.94, 1.0)
NPV	0.959 (0.90, 0.99)	0.980 (0.94, 1.0)
F_1_-score	0.944 (0.88, 0.98)	0.980 (0.95, 1.0)
AUC	0.981 (0.93, 1.0)	0.997 (0.99, 1.0)

**Table 4 cancers-15-02335-t004:** UM/MCC performance 1-stage versus 2-stage (ImageNet + PANDA) training.

	Trained on UM/MCC			
	Benign vs. GS3/4 (1-Stage ImageNet Transfer-Learning)	Benign vs. GS3/4 (2-Stage ImageNet Plus PANDA Transfer-Learn)	GS3 vs. GS4 (1-Stage ImageNet Transfer-Learning)	GS3 vs. GS4 (2-Stage ImageNet Plus PANDA Transfer-Learn)
Accuracy	0.901 (0.79, 0.98)	0.911 (0.81, 0.97)	0.669 (0.53, 0.84)	0.680 (0.54, 0.84)
Sensitivity	0.898 (0.75, 0.97)	0.897 (0.71, 0.97)	0.732 (0.37, 0.93)	0.753 (0.47, 0.90)
Specificity	0.898 (0.75, 0.97)	0.897 (0.71, 0.97)	0.606 (0.26, 0.87)	0.606 (0.19, 0.92)
Precision	0.912 (0.75, 1.0)	0.923 (0.76, 0.99)	0.660 (0.52, 0.86)	0.670 (0.52, 0.90)
NPV	0.912 (0.75, 1.0)	0.923 (0.76, 0.99)	0.699 (0.54, 0.86)	0.712 (0.55, 0.83)
F_1_-score	0.903 (0.78, 0.98)	0.908 (0.77, 0.98)	0.686 (0.47, 0.83)	0.702 (0.51, 0.84)
AUC	0.955 (0.87, 0.99)	0.955 (0.87, 0.99)	0.706 (0.43, 0.90)	0.714 (0.44, 0.90)

**Table 5 cancers-15-02335-t005:** PANDA + UMMCC training.

	Trained on Combined PANDA Radboud + UM/MCC			
	PANDA Radboud Benign vs. GS3/4	UM/MCC Benign vs. GS3/4	PANDA Radboud GS3 vs. GS4	UM/MCC GS3 vs. GS4
Accuracy	0.961 (0.93, 0.99)	0.915 (0.80, 0.97)	0.970 (0.94, 1.0)	0.668 (0.53, 0.84)
Sensitivity	0.944 (0.89, 0.99)	0.902 (0.75, 0.99)	0.971 (0.92, 1.0)	0.647 (0.36, 0.84)
Specificity	0.978 (0.95, 1.0)	0.928 (0.82, 0.98)	0.968 (0.88, 1.0)	0.689 (0.24, 0.87)
Precision	0.977 (0.95, 1.0)	0.927 (0.83, 0.98)	0.970 (0.89, 1.0)	0.687 (0.52, 0.85)
NPV	0.946 (0.90, 0.99)	0.908 (0.78, 0.99)	0.972 (0.92, 1.0)	0.665 (0.55, 0.83)
F_1_-score	0.960 (0.92, 0.99)	0.913 (0.79, 0.97)	0.970 (0.94, 1.0)	0.656 (0.47, 0.84)
AUC	0.988 (0.96, 1.0)	0.963 (0.86, 0.99)	0.996 (0.99, 1.0)	0.710 (0.52, 0.90)

## Data Availability

Available and subject to institutional compliance (data transfer agreement).

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
