# Peer review of "Classifying Malignancy in Prostate Glandular Structures from Biopsy Scans with Deep Learning"

_cancers, 2023, doi:10.3390/cancers15082335_

Round 1

Reviewer 1 Report

Thank your for submitting your work to our journal.

From the beginning I will state that I will focus on the medical part of your paper as I am not qualified to judge on the Deep Learning methodology, which I will presume to be good, for my evaluation.

Apparently your reference manager software got an error, see line 439.

I suggest you review your introduction, maybe reformulate line 35-36 which is very unclear. A brief definition and description of the Gleason score would help here.

You should clarify why you only focused on Gleason grades of 3 and 4, since the scale ranges from 1 to 5 (not widely used in practice but you still need to define your playground).

Although you include a "Limits" chapter in your paper, no clear limit is described. You start this chapter with "another factor..." which is inappropriate.

The conclusions need attention as well. Your aim was to develop a deep learning model to discriminate aggressive Gleason grades and you conclude that your what you got so far is promising. Maybe a more firm conclusion on what works and what the limits are will bring more clarity to your results. "Malignant grade Gleason" should be changed, Gleason is always malignant.

I suggest you have a look and cite the following recent paper: DOI: 10.11152/mu-2832

It focuses on the way biopsy specimens are acquired and dealt with and describes the limits of the biopsies, an important aspect involved in the final description of prostate cancer.

I will gladly review an updated version of your paper.

Reviewer 2 Report

In this paper, authors developed a Deep Learning (DL) model to identify the most prominent Gleason pattern in a highly curated data cohort and validated it in independent dataset. They used transfer learning and fine tuning approaches to compare several deep neural network architectures that were trained on a corpus of camera images (ImageNet) and tuned with histology examples to be context appropriate for histopathological discrimination with small samples. The following review comments are recommended, and the authors are invited to explain and modify.

Comment: “Prostate Cancer Histopathology Classification with CNN Features”, the title does not make much sense.

Comment: Novelty is confusing. A highlight is required. The main contributions of the manuscript are not clear. The main contributions of the ‎article must be very clear and would be better if summarize ‎them into 3-4 points at the ‎end of the introduction.‎

Comment: However, I have several concerns related to the difference with respect to the state-of-the-art and performed experimental results and comparison. It is not clear the improvements with other cited related works.

Comment: Nothing is mentioned about the implementation challenges.

Comment: Methodology is not clear. Provide an algorithm and flowchart of the whole work. The authors need to add a new figure to show the main structure of the proposed system. ‎This will help the reader to get a better understanding of what is going on in the proposed ‎system.‎

Comment: Authors should discuss the stability of the proposed model in terms of complexity.

Comment: This paper does not validate the effectiveness, efficiency, and computing time.

Comment: Moreover, it should be noticed that the clinical appliance has to be decided by medicals since the existing differences between the real image and the one generated by the proposed model could be substantial in the medical field.

Round 2

Reviewer 1 Report

Thank you for making the suggested changes

Reviewer 2 Report

Authors did a good job in paper revision but they need to explain and modify a few concerns before paper acceptance.

1 Introduction section is still weak. An introduction is an important road map for the rest of the paper and should consist of an opening hook to catch the researcher's attention, relevant background studies, and a concrete statement that presents the main argument, but your introduction lacks these fundamentals, especially relevant background studies. This related work is just listed out without comparing the relationship between this paper's model and theirs; only the method flow is introduced at the end, and the principle of the method is not explained. To make soundness of your study must include these latest related works and discuss them.

Recent Advances in Pulse-Coupled Neural Networks with Applications in Image Processing. Electronics, 10.3390/electronics11203264

Automatic interpretation and clinical evaluation for fundus fluorescein angiography images of diabetic retinopathy patients by deep learning. British Journal of Ophthalmology, 10.1136/bjo-2022-321472

Identifying Malignant Breast Ultrasound Images Using ViT-Patch. Applied Sciences, 10.3390/app13063489

2 According to "Figure 1. Prostate Cancer Decision Support System", 2.2 discusses "Image Preprocessing", but where are sections on "Region extraction" and "Feature extraction"?

3. “To train or test on WSIs, (relatively) small image patches were created”, to divide the image into small patches, and then individually predict the class of them. In this way, only the local information of the patch can be used, and the global information of the image is ignored.

4 Could you please check your references carefully? All references must be complete before the acceptance of a manuscript.
